# Courses on Basic Occupational Safety and Health: A Train-the-Trainer Educational Program for Rural Areas of Latin America

**DOI:** 10.3390/ijerph17061842

**Published:** 2020-03-12

**Authors:** Marie Astrid Garrido, Verónica Encina, María Teresa Solis-Soto, Manuel Parra, María Fernanda Bauleo, Claudia Meneses, Katja Radon

**Affiliations:** 1Center for International Health @ Occupational, Social and Environmental Medicine, Hospital of the Ludwig-Maximilians-Universität Munich, Ziemssenstr. 1, 80336 Munich, Germany; 2Instituto de Ciencias de la Salud, Universidad de O’Higgins. Libertador Bernardo O’Higgins 611, Rancagua 2820000, Chile; 3Hospital Provincial del Huasco/Unidad de Salud Mental. Av. Huasco 392, Vallenar 1610000, Región de Atacama, Chile; 4MunBaus Consultores, Av. Mariano Acosta 1254, Buenos Aires 1407, Argentina

**Keywords:** informal sector, community-based participatory research, primary health care, occupational health services, capacity building, social determinants of health, workplace

## Abstract

Integrating basic occupational health services into primary care is encouraged by the Pan American Health Organization. However, concrete initiatives are still scarce. We aimed to develop a training program focusing on prevention of occupational risks for primary healthcare professionals. This train-the-trainer program was piloted at four universities in Chile and Peru. Occupational health or primary healthcare lecturers formed a team with representative(s) of one rural primary healthcare center connected to their university (N_participants_ = 15). Training started with a workshop on participatory diagnosis of working conditions. Once teams had conducted the participatory diagnosis in the rural communities, they designed in a second course an active teaching intervention. The intervention was targeted at the main occupational health problem of the community. After implementation of the intervention, teams evaluated the program. Evaluation results were very positive with an overall score of 9.7 out of 10. Teams reported that the methodology enabled them to visualize hazardous working conditions. They also stated that the training improved their abilities for problem analysis and preventive actions. Aspects like time constraints and difficult geographical access were mentioned as challenges. In summary, addressing occupational health in primary care through targeted training modules is feasible, but long-term health outcomes need to be evaluated.

## 1. Introduction

Globally, work activities in rural areas are mostly linked to the use of natural resources. Examples of such work activities include small-scale mining, fishing, livestock, and agriculture [1]. In agriculture, exposure to hazardous substances, machinery, contact with animals, plants, and vectors is common all over the world [2]. Globally, this results in high morbidity and mortality rates caused by work-related accidents and diseases in rural areas [2]. However, concrete numbers are hardly available partly due to underreporting [2]. Contributing to the poor occupational health outcome, informal employment, temporary employment, unpaid family work, child labor, and the employment of migrants under precarious conditions are common [1,2,3,4,5,6,7].

In addition to these work-related characteristics, poverty, cultural aspects (such as knowledge about and conception of health and illness, language barriers), geographical features, and limited access to health services and to formal education are challenges for rural populations’ health [2,4,8,9]. Within this scenario, a close relationship is established between the absence of social security, poverty, rurality, and lack of access to decent work [1,2]. Furthermore, pollution generated by economic activities can indirectly affect rural populations’ health [10,11,12].

In the framework of the sustainable development goals (SDGs), decent work and good health and well-being are two of the 17 goals. These 17 objectives are closely related and improving one will also have positive effects on the others [13]. Decent work involves “opportunities for work that is productive and delivers a fair income, security in the workplace and social protection for families, better prospects for personal development and social integration, freedom for people to express their concerns, organize and participate in the decisions that affect their lives and equality of opportunity and treatment for all women and men.” [14]. Through this, decent work promotes rural economic growth [2]. Addressing occupational health in rural areas is therefore a necessity [15]. However, only 15% of workers worldwide (and even less in rural areas) have access to at least basic occupational safety and health services (BOSH) [16].

Since its Global Plan of Action on Workers Health for 2008 to 2017, the World Health Organization (WHO) considers the integration of BOSH into primary health care (PHC) as a strategy to universally address workers’ health [1,16,17,18,19,20]. This strategy is supported by the Pan American Health Organization in its Plan of Action on Workers Health 2015–2025 [21]. To the best of our knowledge, concrete actions have only been taken in a few Latin American countries. One example is Brazil, where occupational health was implemented into primary healthcare in the 1980s [22]. However, outcome is still poor, especially in rural areas of Brazil. For example, for one case of pesticide poisoning notified, 50 are not reported [23]. The absence of infrastructure, adequate public policies, accessibility, and trained primary healthcare professionals in occupational health is a prevailing barrier not only in Brazil [1,19,20,24,25].

In many Latin American countries, providing health care services in rural areas is an obligatory part of internships for health care students [26]. However, training on occupational health, prevention, social health determinants, and working with multidisciplinary teams are still scarce for interns [26]. Considering this together with the fact that (1) health promotion and prevention tend to be effective when the community is involved at all stages [27,28,29], (2) community participation is a promising strategy to address work-related health problems [30,31,32,33,34], and (3) implementation of training on occupational safety and health aspects in primary healthcare is encouraged to achieve good health and decent work for rural communities of Latin America [21], we aimed to develop a train-the-trainer program on prevention of occupational risks to be implemented in health-related training at Latin American universities. This article reports on the concept and the evaluation results of the pilot program.

The rest of the paper is divided as follows: We first describe the methodology of the training program. After that, the evaluation strategy is explained. In Section 3 we start by presenting the results of the pilot implementation at four Latin American universities and continue with the evaluation results. In the discussion, we highlight strengths and limitations of the approach in light of the international literature and close with a general outlook and conclusions.

## 2. Materials and Methods

The train-the-trainer program was developed and implemented between January to December 2018. It is based on a blended-learning concept (here: online and onsite training) with a theoretical and practical component. The program was piloted at four universities collaborating with the Center for International Health (CIH). The CIH is a training network on health-related aspects coordinated by the Ludwig-Maximilians-Universität in Munich, Germany. It is funded by the German Federal Ministry for Economic Cooperation and Development [35] “Higher Education Excellence in Development Cooperation” (exceed) initiative. This initiative, coordinated by the German Academic Exchange Service [36], supports the development of competence centers at German universities with their partners in low- and middle-income countries (LMICs), which contribute to the realization of the Sustainable Development Goals (SDGs). In Latin America, the CIH focuses on training in occupational safety and health [37].

From 2015 to 2019, the CIH annually offered parts of its funds as competitive research and training funds for the seven partner universities, associated teachers, and alumni from LMICs. The call was open for any research or training project relevant to the SDGs and in the scope of the CIH’s vision “we empower health professionals”. The project had to be completed within a maximum of 12 months. In 2017, the authors of this paper, together with the four collaborating universities, successfully applied for these funds (number of projects received: 25 of which 6 received funding). The authors of this paper are all lecturers within the CIH training programs in Latin America who jointly designed and carried out the program. In addition, four of them accompanied the participating universities throughout the training program as “tutors”.

The collaborating universities were one CIH partner university and three universities at which CIH alumni are teaching. At all of them, medical or nursing interns provide health services to rural communities. At this stage, we restricted participating universities to two universities located in two geographical regions of Chile (center and central south) and two universities in Peru, one located in the coastal region, the other in the Andean region. This was done in order to limit the travel expenses of the workshops and at the same time, pilot the program in different cultural settings. The universities were informed about the initiative and invited to take part in the training program. All invited universities agreed to participate. Each university was encouraged to participate with a team of lecturers in occupational health and/or PHC and with at least one PHC representative of a rural community (named “team of trainers” herein). The community was selected by each university from those in which the universities’ interns are involved in primary healthcare. The selection was based on the preference of the universities.

### 2.1. Description of the Training Program

The program was taught in two phases (Table 1) and can be downloaded from the Appendix A:

Phase I: Participatory diagnosis of health status and working conditions in rural communities. The objective of this first phase was to provide the teams of trainers with the necessary skills to carry out participatory diagnoses of work-related hazards (i.e., something that may cause damage) and risks (i.e., the likelihood that a hazard causes damage) related to the main production chain in each community following the EcoHealth approach. Nowadays, community participation in healthcare is recommended to customize solutions to local needs [38]. It encourages the community to solve its own problems [39] (p. 177). The EcoHealth approach considers health within its close relationship with environmental and socio-economic factors; therefore, health problems are addressed under its principles of participation, transdisciplinary work, and social, ethnic, and gender equity [40] (pp. 103–147). The methodology was successfully piloted previously by two of the authors (M.A.G., M.P.) in a community of artisanal fishermen in Southern Chile [41]. Using a flipped classroom format, teams were first introduced to the theoretical framework in an online case study (CASUS^®^ [42]). Access to the case study was provided through a Moodle^TM^ platform which was also used as a communication tool (forum, news) and for sharing information and training materials/guides. After this introductory problem-based online training, the four teams met in a three-day face-to-face workshop. In this workshop they learned how to—jointly with the community—use a risk map to visualize the work processes, the related risks/hazards, and the potentially exposed population. In addition, they learned how to jointly search for solutions and resources to address the identified problems [43]. After that, each team, supervised by a tutor, worked on the planning of the participatory diagnosis in the rural communities. After completion of the workshop, all trainers were invited to implement the planned participatory diagnosis in their respective rural communities. During the implementation step they were also accompanied by their tutor.

Phase II: Teaching interventions to address work-related health challenges in the rural communities. Phase II again started with an online introduction followed by a second three-day face-to-face workshop. In this workshop, the teams of trainers got to know active learning methods to prevent work-related health problems and applied them to the problems identified in the community during phase I. “Active learning” means that learning sessions include active application of learning content for all learners in order to raise motivation and improve learning outcome [44]. In order to structure the sessions in such an active way, the ARIPE (adjust, reactivate, inform, process, evaluation) model for competence-based learning was used [45]. According to this model, each training session is divided into five phases and starts with drawing the attention of the learner to the content (e.g., by relating it to an everyday situation (adjust)). After that, the trainer reactivates learners’ previous knowledge on the topic (reactivate). Based on that, the teacher provides the group with new information on the topic (inform), which the learners then process and apply (process). Finally, participants provide the trainer with feedback on their learning experience (evaluate).

Following the workshop, teams of trainers were asked to implement the planned educational intervention in the community. Once more, they were accompanied by their tutor. At the end of the implementation, each team completed a report including feedback from the community members. No (health) outcome evaluation in the communities was done.

### 2.2. Program Evaluation

After implementation of the second project phase in the communities, the trainers completed an online evaluation form (Appendix A). It consisted of 10 questions (Likert scale/close-ended and open-ended questions). Questions aimed at evaluating the five “Attributes of Innovations” that influence their adoption rate as described by Rogers [46] (pp. 10–16):relative advantage, as “the degree to which an innovation is perceived as being better than the idea it supersedes”;compatibility, as “the degree to which an innovation is perceived as consistent with the existing values, past experiences, and needs of potential adopters”;complexity, as “the degree to which an innovation is perceived as relatively difficult to understand and use”;trialability, as “the degree to which an innovation may be experimented with on a limited basis”;observability, as “the degree to which the results of an innovation are visible to others”.

Additionally, a program evaluation workshop was carried out 10 months after the second onsite training due to additional funds received from the Center for International Health. In this workshop, we discussed with the teams:How and to what extent were the program objectives achieved?What facilitated the achievement?Which obstacles were identified in the achievement of the objectives?

Qualitative data of the online survey and the program evaluation workshop were analyzed by two of the authors (V.E., M.A.G.) through the framework analysis methodology. This qualitative research method is suitable to analyze what is happening in a specific setting [47]. Applying this method, the researcher synthesizes the information and obtains common topics; in our case regarding, (1) the attributes of innovation and (2) the quality of the training program.

### 2.3. Ethical Considerations

The universities of the teams of trainers expressed their approval to participate in the course through a letter of commitment. The questionnaire completed by the trainers was anonymous and voluntary. According to the ethical board of the Medical Faculty of the Ludwig–Maximilians-Universität (Munich, Germany) anonymous and voluntary evaluations of training programs do not require ethical approval.

## 3. Results

### 3.1. Participants and Implementation Experiences

At all four universities, occupational health experts (*n* = 5) or PHC lecturers (*n* = 4) underwent the program. Additionally, one vice rector of one of the universities participated. Three universities managed to include at least one PHC representative (total *n* = 5) of the rural community in which they were implementing the project. The rural communities were two fishing villages, one located in the north of Peru, the other located in the central south of Chile; one village dedicated to pottery in the central part of Chile, and finally an agricultural village in the high Andes of Peru. The first workshop was successfully completed by all participating teams of trainers. In the three months following the workshop, all four teams were able to conduct the participatory diagnosis in the communities. The number of community participants in this implementation step varied from five in the Chilean fishing community to 20 in the Peruvian agricultural community (Table 2). The main work-related problems identified were related to musculoskeletal disorders (Chilean pottery and fishing community, Peruvian agricultural community) and mental distress (Peruvian fishing community).

Therefore, teaching interventions developed in phase II focused on prevention of musculoskeletal symptoms (*n* = 3) and work-related stress (*n* = 1). The interventions were successfully carried out in three communities with 11 (Peruvian fishing community) to 68 participants (Chilean pottery community). In the Chilean fishing community, the intervention could not be implemented due to time constraints of the team of trainers and community members.

The verbal feedback trainers received from the communities was positive. Community participants were very satisfied with the participatory and collaborative method, as well as the trustful environment during the activities. They appreciated having the opportunity to learn about risks and hazards and how to take care of their health. They were also motivated to continue these actions involving more members of their communities. Participating community members considered local authorities, educational institutions (i.e., primary school), and PHC workers as important actors for rural workers’ safety and health.

After the end of the project, the three teams which could carry out the interventions continued to work on occupational health aspects with the communities’ PHC centers. In Peru, the type of activities varied from a research project on working conditions of artisanal fishermen to a participatory round table presenting the results and raising awareness among local stakeholders. In Chile, the team started various activities including OSH training of some of the pottery makers so that they would become health promotors and help prevent musculoskeletal problems, OSH training for PHC center staff, as well as coordination of intersectoral activities to raise awareness of occupational safety and health problems in the community. In addition, the PHC professors started implementing parts of the program in the regular training of medical interns before their rural internship.

### 3.2. Quantitative Evaluation Results

Ten of the fifteen trainers completed the evaluation form. The overall evaluation of the training program was 9.7 out of 10. Although trainers suggested to deepen the training, all would recommend this program to their colleagues. According to them, both phases of the program were beneficial to the community compared to traditional, passive ways of learning. Finally, all attributes of innovation, except complexity, showed high mean scores (4+ out of 5) (Figure 1).

### 3.3. Qualitative Evaluation Results

#### 3.3.1. Evaluation Form

Relative advantage

The trainers highlighted that the collaborative work between them and the community showed positive results and facilitated achieving objectives. The simple and participatory methodology provided a respectful environment for approaching the community, overcoming the classic “top-down” knowledge transfer model, as expressed by a trainer:

“I think that with this, local knowledge is respected, validating the work of the people involved, which is complemented by technical knowledge.”(Translated from local language)

Trainers stated that continuity of actions and further interventions are needed to strengthen changes. Impact assessment was also mentioned as being necessary.

2.Compatibility

According to the trainers, the methodology respected the community’s identity, socio-cultural, local political, and gender aspects. A trainer stated: 

“Community beliefs and values were not altered with the implementation of the different stages of the project; the community vision of health-disease process is based on the Andean worldview where mother earth (Pachamama) intervenes as a source of life, gods such as Apus (tutelary hills) that influence the health-disease process … In this Andean society, the woman plays an important role in the family and at the same time, maintains a certain level of dependence on the male…”(Translated from local language)

When implementing the project, sociodemographic characteristics, community knowledge, as well as the participants’ work organization were considered. A trainer said:

“… Considering their work schedules, availability of time and with the participation of assistants of different ages … the educational session was designed (physical exercises) according to health conditions and age … that contribute to improve their ailments without altering their daily work activity.”(Translated from local language)

It became evident that workers’ health problems were related to factors beyond the workplace, which should be considered in the search for solutions. A challenge was seen in how to advance in the process of community participation, in order to increase the level of community empowerment and the collaborative relationship with institutional partners in addressing their issues.

3.Complexity

Complexity experienced by the trainers throughout the project’s implementation was determined by:The isolation and difficult access to the communities due to the geographical remoteness and dispersion of the population within the territory.The political-institutional current situation in Latin America, which made it more difficult to carry out the interventions since in some cases, the local authority approved the intervention and when changed, approval had to be sought again from the successor.Time demand and management for adjusting interventions according to community characteristics, as pointed out by two trainers:

“It took time to define and prepare educational and audio-visual material because of the schooling and age of the participants.”(Translated from local language)

“The organization and assistance by fishermen were complex, 100% was not reached due to work activities of the participants.”(Translated from local language)

Lack of policies in primary healthcare and lack of intersectoral action addressing workers’ health issues. Trainers stated:

“My view of being complex is based on the fact that it requires intense coordination of actions which are not always easy. Primary care does not have a work plan in the area [of occupational health]. Therefore, these efforts imply additional commitments, especially if a greater impact to be achieved”.(Translated from local language)

“I think skills for this kind of work could be strengthened in the healthcare team.”(Translated from local language)

“… I think it is necessary to incorporate follow-up and to develop other instances of intervention, like networking and intersectoral work.”(Translated from local language)

Lack of financial support for teaching interventions (e.g., staff time, travel, and material costs).

Some teams partly overcame the shortcomings as a result of the positive relationship with the community, management, and commitment of some local authorities. A trainer said:

“We had constant support from the university… this allowed us to have sufficient teaching hours, finance travel costs, coffee breaks, materials… Local primary healthcare center contemplated the necessary professional hours, as well as time for coordination of actions with social and institutional organizations.”(Translated from local language)

4.Trialability

Based on their experience, trainers considered applying this methodology to address different health problems and rural communities as feasible. An initial approach to recognizing community health problems and prioritizing more effective measures would be facilitated. Nevertheless, due to the diversity of factors involved in the problems, interventions beyond education should be considered. A trainer mentioned:

“Since the disease model in general has changed, it is important to consider social factors in interventions. Participatory diagnosis and educational intervention models in communities are useful for the development of respectful strategies in which the community is involved.”(Translated from local language)

5.Observability

The local communities’ situation and the internal dynamics of the formed work teams became visible. Through this experience, both the teams of trainers and communities were able to recognize the health problems of the community’s workers, possible solutions, and the obstacles to overcome. Likewise, institutional support facilitated the work of the teams:

“I think we should work with local authorities on a results basis to achieve greater impact.”

“… for the community, the problems arising from the current legislation are more complex …”(Translated from local language)

#### 3.3.2. Program Evaluation Workshop

Ten trainers (representing all four teams) participated in the final evaluation workshop. All declared that the objective of the training program was fulfilled although one of the teams could not complete the teaching intervention. Reasons for this were mainly the geographical distance from the community and difficulties of the team to carry out meetings (time and distance). A trainer mentioned:

“In general, the objectives were achieved, it was possible to make diagnosis of the community’s health problems and their occupational risks. No intervention was carried out.”(Translated from local language)

Factors mentioned to facilitate the achievement of the program’s objectives were related to the motivation of all involved parties as well as the learning methodology. Hindering aspects were time, geography, limited knowledge, and lack of awareness of occupational safety and health, as well as little institutional support (Table 3).

## 4. Discussions

Based on the Pan American Health Organization’s (PAHO) strategy to integrate occupational health services into primary healthcare, we developed and pilot-tested a train-the-trainer program to address occupational risk prevention in rural communities of Latin America. The program was conducted as a combination of online training, face-to-face learning, and implementation practice carried out in two phases: risk assessment and intervention. Community participation was considered at all stages. Overall, evaluation results were positive and implementation was successful in three out of four communities. Three of four teams continued occupational safety and health activities in the communities after the end of the project. One university implemented the program into regular training of medical interns.

As a result of the participatory diagnosis, knowledge was generated regarding the health of workers in rural areas, both for the trainers and for the communities. Musculoskeletal disorders and mental distress were identified as the main work-related health problems and thus, the major rural work-related problems worldwide were targeted [48,49,50]. The teams realized that the relationship between work and community health problems was unknown to the communities’ health care staff. This so-called “blind spot” in occupational health [51,52] of health care providers is well known globally [24,25,41,49,51,52,53,54]. Likewise, local authorities were unaware of the association between work and health. This might indicate that the inclusion of occupational health into PHC as suggested by PAHO still needs to be translated into local policies [21,24].

Joint efforts between communities and teams resulted in teaching interventions based on community needs. Socio-demographic characteristics (e.g., level of schooling, type of work), the use of local languages (i.e., Aymara, Quechua), and the low educational level of many community members were challenging. However, for the successful implementation of the program it was important to consider them as it is known that innovations become more widely adapted when they are socio-culturally accepted [46] (p. 15), and trust exists between community members and health care providers [55].

Our program raised awareness about the association between work and health and the prevention of occupational risks in the community. Previous studies suggest that the community-based approach taken contributed to the success of the program [29,31,33,56]. PHC should be involved given its close relationship with the communities [57] and in order to ensure the sustainability of further actions. In our case, the trainers’ teams worked on subsequent interventions, enabling progress in improving working conditions and providing tools for community empowerment. At one of the universities, the program could already be implemented into the regular preparation of medical students for their rural internship. Such a regular implementation would be the long-term goal of the program once its effectiveness on community health was shown. It is interesting to note that at this especially successful university the institutional support was high. In addition, the team was rewarded by the university for their successful work with the community through a travel grant (personal communication). On the contrary, in one community the planned intervention could not be implemented. Reported reasons were related to the difficulty in adapting the trainers’ times to those of the community members. This involved the geographical conditions of the community which required long commuting times. It is interesting to note that in this community no community member could be involved in the team of trainers. In addition, occupational health lecturers at this university were not full-time professors and no extra hours were assigned by universities’ management for the program. This highlights the importance of community involvement from the beginning as well as the importance of institutional support in programs like this [58].

Lack of public policies, missing involvement of authorities, and lack of intersectoral action became the main factors of complexity. According to the International Labour Organization (ILO), low social security coverage is a major shortcoming in terms of access to decent work in rural areas; cross-ministerial policies encompassing the scope of occupational safety and health in rural development are needed [1]. Some countries [49,59] have already implemented in their public policies the provision of occupational health services in primary care; however, the lack of trained health care providers in occupational health is a global challenge [60]. This need was also perceived by the trainers in our project and therefore, our training program was well received. Teams even suggested to deepen the training; however, longer workshops would have been difficult to carry out due to time constraints especially for the rural health care professionals. One has to bear in mind, that frequently only one health care professional works together with an internal in a rural community and thus, leaving the community for longer periods of time is hard to compensate [25]. Therefore, we offered parts of the program online. While those who completed the virtual part indicated the benefits of it, only 6 out of the 15 participants completed all online parts of the program (data not shown). However, as we had to rely on compliance of our participants, this figure indicates their high interest in the program. In comparison, less than 1% of medical students completed e-learning components of the medical curriculum when no external motivation was provided [61].

As a limitation, the methodologies and strategies used might not be suitable in all parts of rural Latin America. However, the multi-national team of authors of the program (Chile, Bolivia, Guatemala, Argentina), the involvement of trainers’ teams from northern Peru to the central south of Chile, as well as the flexibility of the used methods are promising for its implementation in other contexts. The project duration was also a limitation, especially for trainers’ teams as they had to organize the workshops in the communities. Likewise, the duration of the project, limited financial resources, and lack of experience with research of most of the participating teams did not allow evaluation of the impact of the actions on communities’ health. In addition, no control group was included. Furthermore, not all trainers completed the evaluation form or took part in the final workshop. Although all participants were invited, participation was voluntary, respecting the personal decisions of the trainers and their availability of time. While we cannot know to which degree the written evaluation was representative, members of all four teams took part in the final evaluation workshop. Therefore, we expect selection bias to be small. Reporting bias might have played a role especially in the final workshop; however, as there was an atmosphere of trust and respect built over the training program, we do believe that this had only a minor impact on the results.

## 5. Conclusions

We have successfully developed a train-the-trainer program on occupational risk prevention and have shown the feasibility of its implementation in rural areas of Latin America. The cornerstones are its participatory community-based approach and active learning methodology combining classroom teaching with guided practice. In the next step, we need to test its medium- and long-term impact on communities’ health. After successful evaluation, it can be used as a tool to implement occupational safety and health in rural areas. For this, we consider cooperation with relevant intersectoral stakeholders and institutional support to be crucial.

## Figures and Tables

**Figure 1 ijerph-17-01842-f001:**
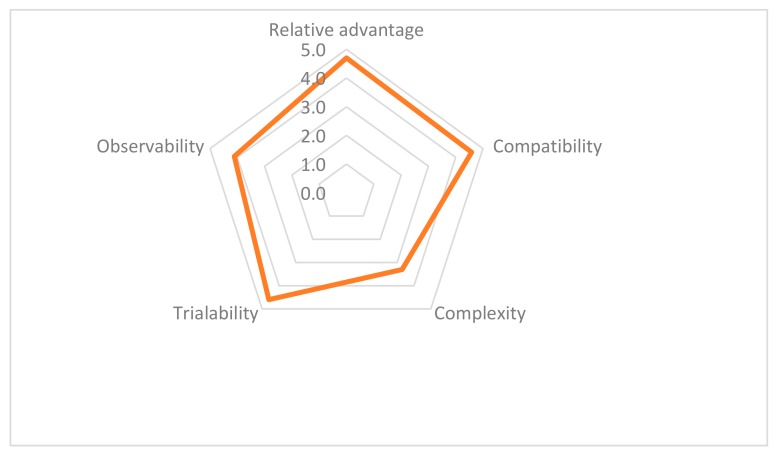
Score for each attribute, based on the mean of five Likert questions rated from 1 to 5.

**Table 1 ijerph-17-01842-t001:** Activities, topics, content, and schedule of the training program on occupational risk prevention in rural communities.

Phase	Activity	Topic	Content/Activities for the Trainers	Time Period
Participatory diagnosis	1-h problem-based e-learning for trainers	Work-related hazards and risks in rural areas	Basic information on: Common work-related hazards in rural areas with potentially adverse effects on workers’ and families’ health Community actors who should be involved in risk control Preventive measures	March–April 2018
	Three-day face-to-face workshop for trainers	Participatory diagnosis of working conditions and health	Intercultural communication Approaching the community through the work–health relationship Introduction to participatory diagnosis Risk mapping as a tool of participatory diagnosis Identification of work processes in the community Hazards identification Prioritization and analysis of one problem using the following questions based on the EcoHealth approach [40]: What is the problematic situation? What is the potential damage to health? How does the problem affect different community members? Who knows about the problem? Who should be involved in addressing the problem? Which is the possible source of the problem?	April 2018
	One-day workshop for community members offered by trainers	Implementation of the participatory diagnosis of working conditions and health	Organization of the workshop and invitation of community members Diagnosis of working conditions and health in the community Selection of the most relevant work-related health problem to be addressed in the teaching intervention	May–July 2018
Teaching intervention	2-h problem-based e-learning for trainers	Introduction to teaching interventions	Problem tree analysis Scientific literature search	July–August 2018
	Three-day face-to-face workshop for trainers	Teaching interventions to address work-related health challenges in rural communities	Introduction to teaching interventions Drafting the concept of the teaching intervention Background Context (framework conditions) Learning objectives Active learning based on the ARIPE Model (adjust, reactivate, inform, process, evaluate) [45] Planning the teaching intervention Practice and feedback Evaluation	August 2018
	Half-day workshop for community members offered by trainers	Implementation of the teaching interventions	Organization of the workshop and invitation of community members Teaching intervention workshop to prevent the most relevant work-related health problem identified with the community	September–November 2018
Evaluation	Online questionnaire for trainers			November 2018
	Evaluation workshop for trainers			September 2019

**Table 2 ijerph-17-01842-t002:** Findings from the participatory diagnosis, topics of the teaching interventions, and related activities carried out by the teams of trainers.

Trainers’ Teams	Communities/ Country	Phase I: Participatory Diagnosis		Phase II: Teaching Intervention		Further Activities
		Community Participants	Work-Related Risks and Resulting Health Problems Identified	Community Participants	Content	
1 ^1^	Artisanal fishermen/Peru	N = 6 (3 women) Representatives of Health care workers (*n* = 2) Fishermen (*n* = 2) Community health promotors (*n* = 1) Municipality (*n* = 1)	Psychosocial risks (work demands and interpersonal relationship) Ergonomics risks (overstrain) Environmental pollution (transport)	N = 11 men (0 women)Representatives of: Fishermen (*n* = 11)	Self-control of mental distress in artisanal fishermen	Planning a study on working conditions of artisanal fishermen
2 ^2^	Potato and quinoa farmers/Peru	N = 20 (6 women) Representatives of: Community members (*n* = 10) Primary health care workers (*n* = 3) Primary school teacher (*n* = 3) Community leaders (*n* = 3) Local authorities (*n* = 1)	Ergonomic risks (load handling, repetitive movements) resulting in lower back pain	N = 26 (13 women)Representatives of: Community members (*n* = 17) Social health insurance (*n* = 1) University (*n* = 1) Regional health officer (*n* = 2) Primary school teachers (*n* = 5)	Prevention of lower back musculoskeletal disorders	Participatory round table to present the project results and to raise awareness among decision makers about health of rural workers
3 ^3^	Pottery makers/Chile	N = 17 (14 women)Representatives of: Pottery makers (*n* = 12) Medical students in their internship (*n* = 2) Primary healthcare authorities (*n* = 2) University administrator (*n* = 1)	Ergonomic risks (repetitive movements, inadequate posture, load handling) resulting in musculoskeletal problems Psychosocial risks (presentism, double burden, work schedule) resulting in mental distress	Session 1: N = 16 (14 women)Representatives of: Female pottery makers (*n* = 14) Medical students in their internship (*n* = 1) University administrator (*n* = 1) Session 2: N = 52 female pottery makers	Prevention of musculoskeletal disorders	Since 2018, best practices in health projects including: Training of community health promotors to prevent musculoskeletal disorders Training of primary health care center staff Interventions to improve working conditions for pottery makers Intersectoral coordination
4 ^4^	Artisanal fishermen/Chile	N = 5 (2 women) Representatives of: Fishermen/shore collectors and relatives (*n* = 5)	Ergonomic risks (poor posture, load handling) resulting in lower back pain Biological risks resulting in infectious diseases Physical risks resulting in decompression illness	Not implemented due to time constraints of the team of trainers and community members	Prevention of musculoskeletal disorders	None

^1^ Team composed of one university occupational safety and health (OSH) expert and three community primary health care (PHC) representatives, all of them men in the age range 40 to 44 yrs. Tutor was a female OSH expert from Chile. ^2^ Team composed of two university OSH experts (one man, one woman) and one male university representative; age range 20 to 69 yrs. Tutor was a female OSH expert from Argentina. ^3^ Team composed of two university PHC lecturers and two primary health care representatives, two men, two women in the age range of 30 to 59 yrs. Tutor was a female OSH expert from Bolivia. ^4^ Team composed of two university OSH experts and two PHC lecturers, three men, one woman in the age range 30 to 44 yrs. Tutor was a female OSH expert from Chile.

**Table 3 ijerph-17-01842-t003:** Factors identified by the trainers as facilitators and hinderers for achieving the objectives of the program.

Facilitating Factors	Situations to Which These Factors Applied
Commitment and involvement: trainer teams, communities, local authorities, and universities	Prior to the participatory diagnosis and teaching intervention sessions, the teams met with local authorities, workers’ associations, and other community actors, achieving commitment and support for the activities. As an example, in two of the communities the participatory diagnosis and teaching intervention took place in the primary school. Likewise, teachers collaborated in convening the workers, as most of them were parents of school children. Some universities, local authorities, and healthcare centers provided financial support for transportation, teaching materials, and catering.
Working closely with the community	The pre-existing formal relationship between universities, healthcare centers, and communities contributed to the inclusion and sustainability of the activities during the training program. Respect, trust, and empathy facilitated finding common ground in participatory diagnosis and searching for solutions.
Participatory methodology	The role of facilitators assumed by the trainers along with involvement of diverse community actors promoted the identification of work-related health problems in the community. Knowledge from community members and trainers was mutually recognized and shared (e.g., community workers and schoolteachers were key in building the risk map). Teachers were knowledgeable about the work of the community but unaware of the risks involved. They were also helpful in adapting teaching materials to the local language. Some trainers were aware of funding sources for the sustainability of actions with the community.
Flexibility of the methodology	Although the participatory diagnosis and teaching intervention phases were pre-structured, the teams could adapt the methods to meet their own and the communities’ needs. Examples of these adaptations included: Flexible duration of the sessions. Learning methods and materials adapted to age, educational level, and language of the community. Before carrying out the participatory diagnosis, some teams conducted field trips to learn about the main work activities in the community.
Position of the trainer teams at the university	Being a full-time professor and having a good relationship to university management facilitated the involvement of lecturers in the program.
Availability of time for the activity	Some universities assigned hours of the teaching staff for the program. This enhanced commitment of the trainers and smoothed adjustment to community participants’ schedules.
Coordinated teamwork	Respect, communication, and commitment favored teamwork, especially when trainer teams were composed of lecturers from different departments.
**Hindering factors**	Situations to which these factors applied and potential solutions.
Lack of knowledge on occupational health issues	Trainers and community participants involved in primary health care expressed the need to deepen occupational health knowledge. The lack of knowledge was overcome thanks to the active learning methods, mentorship, and teamwork with occupational safety and health experts.
Low level of schooling of the community participants	Low level of schooling required adaptation of the learning material. Active learning methods helped to master this challenge.
Lack of time for the activity	In some cases, the trainers’ workload together with the irregular work schedule of community members made it difficult to organize activities. Including the program as part of institutional activities allowed trainers to allocate time for the program.
Geographical distance and accessibility of the communities	Some communities were hard to reach due to their geographical isolation resulting in long commuting times for the teams of trainers. Careful arrangement of schedules and sufficient time allocated to the activities are necessary for the successful implementation of the program.
Limited economic resources	Funding was needed to implement the participatory diagnosis and teaching intervention in the community (e.g., transport, catering, teaching materials). The teams managed to receive financial support from the universities and stakeholders (e.g., in one case the regional health office provided the catering).
“Hiddenness” of work-related illnesses	Lack of awareness about the association between work and health lowered the priority given to occupational safety and health activities. Health care workers realized that work-related health problems were not visible in the primary healthcare centers prior to the activity. For some workers of the communities it was the first time they understood the relationship between hazardous working conditions and their health problems. Beliefs such as “the man in the countryside can tolerate more effort than the man in the city”, as stated by one team of trainers, influenced the conception of occupational health. Implementing the program in the communities changed their perspectives.
Lack of institutional support	One trainer perceived a lack of institutional support from the university as one factor hindering the achievement of the program’s objectives. Institutional support and rewards for engagement in the program need to be considered from the beginning.
Lack of local authorities’ awareness about the relation between work and health	Lack of awareness about the importance of work for health required additional efforts to involve local authorities. Careful prior information provided in for example, regional round tables for local authorities, raises awareness and helps receive support.
Local political situation	The planning and implementation of activities were affected in two communities by a change of regional authorities, requiring seeking approval twice, and a strike of the workers of their target group. Unforeseen circumstances need to be considered when planning the activities.

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
