# Peer review of "Courses on Basic Occupational Safety and Health: A Train-the-Trainer Educational Program for Rural Areas of Latin America"

_ijerph, 2020, doi:10.3390/ijerph17061842_

Round 1

Reviewer 1 Report

This is an excellent and sorely needed study.  It provides a rigorous example of evaluation of OSH training, and the community approach is exemplary.  It also was likely very complicated to execute.

Reviewer 2 Report

General remarks

The paper deals with a very important and interesting topic on occupational safety and health in rural Latin America. This is an area that has received little scholarly attention, especially as it relates to rural communities in non-Western countries. That said, a few things need tightening though to further enhance the quality of the paper.

Abstract

The abstract contains unnecessary information. The abstract needs to be concise and should be rewritten in order to make it attractive to readers (simple sentences without any repetition) and include 2-3 sentences ready to be cited exactly as they are. In one paragraph, your abstract should (a) tell the readers why the study is important (maximum 25% of the text), (b) what you did, i.e. your methodology (maximum 25% of the text), and (c) what you found, i.e. main research results and their major implications (50% of the text). This is very important to promote your work because of the growing trend that authors use Google search to find and cite papers based on the abstract (instead of reading the full paper).

Also, the first sentence ‘Precarious employment is common in rural areas’ (line 18) should be deleted, as it is unrelated to the other sentences.

Introduction

-The introduction is well-written and comprehensively referenced to the literature and identifies a research gap which the authors seek to fill.

-The authors refer to ‘decent work’ repeatedly, but its conceptualization is non-existent.

-The authors should insert a paragraph after line 78 that states how the paper is organized, as this will guide the readers. For instance, ‘the rest of the paper is divided as follows: the next section outlines the material and methods...’ In other words, this should be the last paragraph before you move to the next section (that is, ‘Materials and Methods’).

Conclusion

-Although this is an interesting paper, the conclusions are lacking analytical depth and theoretical implications.

-Additionally, it is hardly path breaking to conclude in this journal that participatory methodologies are useful for teaching basic occupational health for the diagnosis and approach of workers’ health in rural areas through primary care. The authors should highlight the study’s notable findings.

Reviewer 3 Report

General comments

The Manuscript „Courses on Basic Occupational Safety and Health: A train-the-trainer educational program for rural areas of Latin America“ addresses the problem of the lack of occupational health services in rural areas by proposing a program which was implemented and evaluated among Latin American trainers and rural ares.

Although the Manuscript is interesting, it is difficult to understand what kind of the study this is. There is an intervention, but the effect of the intervention is not measured or just not reported. There is a training, and the trained persons discuss and evaluate the training they have had, but there is no actual comparison to any other approach.

In general, I believe that the topic is important and of interest, and that the Manuscript can be improved to provide an interesting perspective on the issue of BOHS in Latin America, but a lot of work is needed by the Authors.

Please take a refer at the attached PDF for specific comments in various sections!

Abstract

The abstract is unecessarily vague and offers general information without stating who performed the training, where exactly the training occured, in what community was the program implemented, etc.

The first phrase of the Abstract talks about precarious work, but this kind of work is not mentioned later in the abstract – if the authors have addressed precarious work specifically, this should be addressed further by the abstract.

In addition, it is unclear what access precarious workers have even to primary healthcare services, if any.

The Methods section of the Abstract is extremely vague, it is unclear who trained whom, where they came from, what happened afterwards, where was this later implemented in the community.

The abstract should give the readers much more information, and here it is like a teaser. The abstract should allow the reader to understand what the Authors did even without reading the full text.

Introduction

The Introduction starts with vague and general information. The Authors should concentrate on the actual information existing for their area of interest (Latin America) or state if the information does not exist. Specific figures are needed to help the reader understand why the problem is being addressed.

The references (first paragraph) are mostly 10+ years old. Has there been no research on rural areas, work, work-related accidents, etc?

The Authors are citing (relatively old) literature but no research regarding Latin America. This seems strange. A completely new and thorough literature search should be done regarding (basic) occupational health services in rural areas, and the Authors should try to synthetyze the knowledge in order to justify their attempt. It feels as if a whole section in the Introduction regarding basic occupational health services in rural areas (studies, published papers) is missing. The Authors should try to find studies about this topic, and other topics related to community trainers, programs, interventions, or be very clear that studies do not exist.

The interest of WHO in BOHS is reported, but using a 10 year old reference. In case the topic of BOHS is still an interest of the WHO the Authors should try to show it.

Another paragraph concerning some of the later used approaches, such as community diagnosis, participatory diagnosis, cultural differences and general special characteristics of rural Latin America are unknown to the reader and should be addressed (and not only discovered later in Table 2).

Methods

Similarly to the Introduction, the Methods section is also vague. No information is given about the Centre organizing the training, nor about the „four universities“ from which the trainers were recruited. How were they selected, how were they invited, who applied (response rate)?

The Authors must provide more information about the training program – just the timetable is not enough. I propose a simple table format of two columns, on the left the TOPIC, and on the right the main contents for this topic. Authors should think about reproducibility of their study, it is necessary to understand the training program.

Rural communities of the respective trainers are mentioned several times, but no specific definition, location or additional information is provided. The Authors should provide this information before starting to use rural communities as a target.

The Evaluation methodology requires additional paragraphs of explanation. Please explain any underlying methodology your evaluation is based on so there is no need to read the cited reference to understand the basis of your approach. Also, please explain how were the participants invited to participate in the evaluation, how was the questionnaire delivered and filled in (pdf, word, google forms, another system).

The S1 and S2 should be cited more precisely, stating that the information is available to be downloaded as Supplementary material.

Results

The results start with some general numbers regarding programme participants (trainers) but do not share the information regarding the country, university, education, age, etc. of the trainers.

The Authors should try to organize Table 1 in a better way. Data should be shown in an uniform way. It is unclear what time constraints prevented the last team from doing their community intervention.

The authors should try to provide a timeline, maybe graphically, clearly showing different activities from the recruitment to the end of the evaluation of the project with participants, communities, their characteristics, problems, etc... This could substitute Table 1 if the Authors see fit.

Since the topics and their depths is unclear from the Manuscript, it is also unclear what is ment with „participants wanted to deepen the topics discussed“. The Authors should provide more information.

Finally, from Line 210 to Line 214 the Authors provide the level of detail necessary for the understanding of their study. Explanations of this kind are lacking in the Introduction and in the explanation of the approach, and would be beneficial for the reader.

All of the proposed interventions need a paragraph explaining what was done. In case there is a pre-post evaluation of symptoms it should be stated. In case this was not done, it should be stated.

It is unclear why did not all participants participate in the evaluation. The Authors should explain.

Was evaluation offered also the the community participants? How was the efficacy of interventions measured?

Table 2 would be much more useful if the Authors identified first if a specific point refers to trainers or community participants. Then a practical example for each of the points is necessary – how was the flexibility of the methodology applied/how did it help and in what situation.

Discussion and conclusions

The first paragraph of the Discussion should be rewritten to summarize the need, the approach, and the general results of this study. Currently, the phrases seem very general and unrelated one to another.

Several terms showing up in the Discussion (and other sections) should be explained and addressed in the Introduction. I have marked all those terms in the attached PDF file.

The Authors should put an additional effort to discuss (using published literature) some of the most important and interesting findings of their study/approach: e.g. the idea of the workers and the community health personnel that various OH diseases or symptoms are „normal“ and should not be addressed.

Unfortunately, the Authors mention several things in the Discussion for the first time – such as the educational level of the community workers. The information which was not presented in the Results cannot be discussed.

Lines 330-332: Please do not cite literature without explaining and clearly stating what was your results and what from the cited reference you are comparing it to.

Round 2

Reviewer 2 Report

The authors have addressed my concerns, so I recommend the paper for publication.

Reviewer 3 Report

The Authors have put an effort and have significantly improved their paper. Therefore, I have no reason not to recommend publication.